# Protein Secondary Structure Affects Glycan Clustering in Native Mass Spectrometry

**DOI:** 10.3390/life11060554

**Published:** 2021-06-11

**Authors:** Hao Yan, Julia Lockhauserbäumer, Gergo Peter Szekeres, Alvaro Mallagaray, Robert Creutznacher, Stefan Taube, Thomas Peters, Kevin Pagel, Charlotte Uetrecht

**Affiliations:** 1Leibniz Institute for Experimental Virology (HPI), 20251 Hamburg, Germany; hao.yan@leibniz-hpi.de (H.Y.); julia.lockhauserbaeumer@leibniz-hpi.de (J.L.); 2Organic Chemistry, Free University Berlin, 14195 Berlin, Germany; gergo.peter.szekeres@fhi-berlin.mpg.de (G.P.S.); kevin.pagel@fu-berlin.de (K.P.); 3Fritz-Haber-Institut der Max-Planck-Gesellschaft, 14195 Berlin, Germany; 4Center of Structural and Cell Biology in Medicine (CSCM), Institute of Chemistry and Metabolomics, University of Lübeck, 23562 Lübeck, Germany; alvaro.mallagaraydebenito@uni-luebeck.de (A.M.); r.creutznacher@uni-luebeck.de (R.C.); thomas.peters@chemie.uni-luebeck.de (T.P.); 5Institute of Virology, University of Lübeck, 23562 Lübeck, Germany; stefan.taube@uni-luebeck.de; 6European XFEL GmbH, 22869 Schenefeld, Germany; 7Centre for Structural Systems Biology (CSSB), 22607 Hamburg, Germany

**Keywords:** ion mobility, native MS, electrospray ionization, norovirus capsid protein, carbohydrate binding, HBGA

## Abstract

Infection by the human noroviruses (hNoV), for the vast majority of strains, requires attachment of the viral capsid to histo blood group antigens (HBGAs). The HBGA-binding pocket is formed by dimers of the protruding domain (P dimers) of the capsid protein VP1. Several studies have focused on HBGA binding to P dimers, reporting binding affinities and stoichiometries. However, nuclear magnetic resonance spectroscopy (NMR) and native mass spectrometry (MS) analyses yielded incongruent dissociation constants (K_D_) for the binding of HBGAs to P dimers and, in some cases, disagreed on whether glycans bind at all. We hypothesized that glycan clustering during electrospray ionization in native MS critically depends on the physicochemical properties of the protein studied. It follows that the choice of a reference protein is crucial. We analysed carbohydrate clustering using various P dimers and eight non-glycan binding proteins serving as possible references. Data from native and ion mobility MS indicate that the mass fraction of β-sheets has a strong influence on the degree of glycan clustering. Therefore, the determination of specific glycan binding affinities from native MS must be interpreted cautiously.

## 1. Introduction

Human norovirus (hNoV) infection is the most common cause of acute gastroenteritis, leading to an estimated 685 million cases annually worldwide. The elderly, immunocompromised patients and children under 5 years are the most severely affected. hNoV belongs to the family of *Caliciviridae*, non-enveloped viruses of icosahedral shape with the viral genome consisting of positive-sense single-stranded RNA. Histo blood group antigens (HBGAs) serve as attachment factors in viral infection [1,2]. Previous nuclear magnetic resonance spectroscopy (NMR), X-ray crystal structure and mass spectrometry (MS) investigations identified the L-fucose moieties within HBGAs as a minimal binding motif for hNoV attachment. Most studies use the dimers of the protruding domain (P dimers) of the major capsid protein VP1. Notably, L-galactose derived from L-fucose by substituting one hydrogen atom for a hydroxyl group at C6 is known not to bind [3,4]. Because the determination of dissociation constants and binding stoichiometries is key to understanding protein–ligand interactions, a number of studies reported such data for the binding of HBGAs to hNoVs based on different experimental approaches. Sun et al. previously established an MS-based method to determine specific glycan binding to proteins [5], which relies on the addition of reference proteins to the sample mixture and simultaneously analysing the constituents using native MS. This approach allows the quantitation and elimination of unspecific ligand clustering during electrospray ionization (ESI). Clustering arises from the statistical presence of free ligands within the same droplet as the free or ligand-bound proteins, which can then dry down to the protein surface upon droplet evaporation. This process becomes relevant at elevated ligand concentrations and is independent of protein molecular weight [6,7,8]. Calculations to correct for this effect are based on total peak areas per mass species within the same spectrum to ensure identical ionization conditions.

A couple of groups including ours have employed this method to characterize glycan binding to hNoV P dimers using distinct reference proteins [4,9,10,11,12]. Notably, results disagreed [4,11], raising the question of whether the selection of a reference protein influences data interpretation. Moreover, with the exception of glycan mimetics [13], the search for non-binding ligand or P dimer controls for native MS was unsuccessful in our hands. This is in stark contrast to NMR data, which show that certain P dimers and glycans do not interact. This suggests a severe issue with the native MS approach. Strikingly, direct MS analysis determined mM K_D_s for binding of several sialic acid containing carbohydrates hNoV P dimers [9,11,12,14], whereas orthogonal saturation transfer difference (STD) and protein-based chemical shift perturbation (CSP) NMR experiments clearly revealed no binding of sialic acids to hNoV P dimers or virus-like particles (VLPs) [3,9,12]. This questions the validity of the results from direct MS measurements.

Additionally, the reported K_D_ values from carbohydrate- binding studies to P dimers are not comparable in STD NMR, native MS and isothermal titration calorimetry (ITC) [11,12,15]. Importantly, K_D_s obtained for active pharmaceutical agents based on different biophysical assays such as native MS, ITC, surface plasmon resonance (SPR) and circular dichroism (CD) were equivalent in numerous other cases [8,15,16]. However, these compounds showed higher binding affinities (µM range) for the target protein and exhibited in most cases no carbohydrate-like structures [17]. In contrast, glycan binding affinity is often in the high µM to mM range. This also holds for hNoV–carbohydrate interactions [9,10,11,12,14,15]. In former studies, the origin of the discrepancies between NMR and native MS data for P dimer–glycan interaction was not in focus [12]. The problem became evident when comparing non-deamidated (wildtype), and deamidated GII.4 Saga P dimers. In this GII.4 P dimer, an asparagine residue flanking the binding site is specifically and spontaneously converted into an iso-aspartate. The deamidated P dimer has been shown to have greatly reduced glycan binding affinity in NMR and hydrogen/deuterium exchange (HDX) MS, roughly by an order of magnitude for HBGA B trisaccharide and fucose compared to the wildtype [4,13].

Here, wildtype and deamidated P dimers are compared using native MS to shed light on potential methodological issues. Moreover, Gb4, an all-galactose tetrasaccharide, is employed as a negative control. Using this information, we hypothesize that clustering depends on physicochemical properties of the proteins. To deduce what obscures the binding studies, multiple reference proteins varying in properties are compared. The results point to an influence of β-sheet content. This theory is corroborated by ion mobility MS (IMMS) measurements on P dimers in presence of glycans and additional data on other P dimers (e.g., from MNV P dimers), which were recently shown by NMR not to bind glycans at all [18]. Our results indicate that reference proteins need to be chosen carefully to match the structural properties of the target protein for glycan binding studies, and, crucially, they suggest the additional influence of structural dynamics that preclude glycan-binding studies in native MS for hNoVs.

## 2. Materials and Methods

### 2.1. Glycans

The following glycans were purchased from Elicityl-Oligotech, dissolved in H_2_O for native MS analysis and are shown in Scheme 1. Blood group antigens: (1.) A tetrasaccharide type 1 (>90% NMR) (GalNAcα1-3(Fucα-2)Galβ1-3GlcNAc, product code: GLY035-1-90%), (2.) B tetrasaccharide type 1 (>90% NMR) (Galα1-3(Fucα1-2)Galβ1-3GlcNAc, product code: GLY038-1-90%). Globo-series: (3.) P antigen Gb4 (>90% NMR) (GalNAc β1-3 Galα1-4Galβ1-4Glc, product code: GLY121-90%). The HBGA ligands were chosen as known binders based on their fucose binding moiety, whereas Gb4 was the fucose-free non-glycan-binding reference. HBGAs comprise various oligosaccharides, the tetrasaccharides were chosen here instead of e.g., trisaccharides to have a larger mass increment upon association with the P dimer to allow for gentler conditions in the native MS measurements.

### 2.2. Proteins

P dimers of hNoV VP1 were used as target proteins. Native MS investigations were performed with P dimers from the hNoV strains GII.4 Saga (Saga 2006 (GenBank ID: AB447457, aa 225–530), GII.4 MI001 (KC631814, aa 225–530) and murine noroviruses (MNV-1, CW1: DQ285629, aa 228–530). *Escherichia coli* was used for overexpression of the P domains as described in [4,18]. Purified P dimers were stored at 4 °C before preparation for native MS experiments. The commercially available reference candidates cytochrome c from equine heart (cyt c, CAS number 9007-43-6), ubiquitin from bovine erythrocytes (Ubq, CAS number 75986-22-4), carbonic anhydrase isozyme II from bovine erythrocytes (CA, CAS number 9001-03-0), alcohol dehydrogenase from *Saccharomyces cerevisiae* (ADH, CAS number 9031-72-5), L-lactic dehydrogenase from rabbit muscle (LDH, CAS number 9001-60-9), myoglobin from equine heart (Myo, CAS number 100684-32-0), human apo-transferrin (apo-TFF, CAS number 11096-37-0) were purchased from Sigma-Aldrich (Merck, Germany) and stored according to manufacturer recommendation. Furthermore, a superfolder green fluorescent protein (GFP) was kindly provided by Henning Tidow (University of Hamburg, Hamburg, Germany) and used as a reference protein candidate [19].

### 2.3. Native Mass Spectrometry

Prior to MS analysis, purified P dimers were buffer exchanged to 150 mM ammonium acetate (Sigma-Aldrich, reagent grade ≥98%) at pH 7 via centrifugal filter units at 12,000× *g* at 4 °C (Vivaspin 500, MWCO 10000, Sartorius). Glycans for native MS analysis were mixed with 1 µM (per monomer) purified GII.4 Saga P dimer and 3 µM of the reference proteins at indicated concentrations. Native mass spectra were measured at room temperature in positive ion mode on a high mass modified LCT Premier mass spectrometer (Waters/Micromass, UK and MS Vision, the Netherlands) with a nano-ESI source [20]. In house-made gold-coated electrospray capillaries were used for direct-sample infusion without any accessory chromatographic separation. The voltages and pressures were optimized for non-covalent protein complexes. The gas pressure used in the source hexapole was set between 6.5 and 6.8 × 10^−2^ mbar argon optimized for minimal complex dissociation and the backing pressure of the source roughing pump was set between 7.0 and 8.5 mbar. Spectra were recorded with applied voltages for the capillary and cone of 1.20 kV and 240 V, respectively. Note that charge state distributions for P dimers were sometimes slightly shifted as was the degree of glycan clustering per charge state. These changes were highly variable and not quantifiable. We attribute them to spray variations and hence slight variation in activation.

For calibration of the raw data, a 25 mg/mL cesium iodide spectrum from the same day was used. MassLynx V4.1 (Waters, Manchester, UK) was used to assign peak series to protein species and to determine the mass after minimal smoothing. Furthermore, OriginPro 2016 SR2 (OriginLab, Northampton, MA, USA) was used for spectral deconvolution and curve fitting. Correction for nonspecific protein-ligand clustering was performed as described [5]. Corrected peak areas were averaged and normalized to the free-protein signal. Data were based on at least three independent measurements. Python code Panda and Numpy package were used to calculate the specific binding of P dimers according to the glycan clustering ratio obtained from the selected reference protein candidates. Seaborn and matplotlib.pyplot package were imported into a python script to plot the results. The module of sklearn.linear model was applied for linear regression. The matplotlib and heatmap package were used to plot the heatmap.

### 2.4. Determination of Glycan Binding: Calculation of the Dissociation Constant (K_D_)

Data from native ESI MS measurements were translated into dissociation constants (*K_D_* values). The detailed calculations were based on the equations listed below. A protein (*P*) and a glycan ligand (*L*) form a complex through non-covalent interactions leading to a reversible association described by (Equation (1)):(1)P+L⇌PL
when equilibrium is reached, the *K_D_* can be calculated directly from the concentrations via the law of mass action (Equation (2)):(2)$$K_{D}=[P_{free}]\left[L_{free}\right]/\left[PL\right]$$

Theoretically, the *K_D_* is calculated by measuring the concentration of the three components (*P_free_*, *L_free_*, *PL*) in solution. Usually, these values are only indirectly accessible because they require ligand titration up to binding saturation. However, native MS allows for the directly measurement of *P_free_* and *PL*, from which *L_free_* can be calculated using the input concentrations. In the present work, the non-covalent interaction was measured in the gas phase assuming identical ESI and detection efficiency for free and glycan-associated proteins, both for the analyte and the reference protein. The peak areas (*A*) of the non-bound protein ion (*P^n+^*) and the bound protein ion (*PL^n+^*) were used to calculate the ratio (*R*) as these most accurately reflected the total signal and hence were assumed to be comparable to the ratio of a ligand-bound protein complex and an unbound protein in solution (e.g., equilibrium state). For P dimers, the equations reflect glycan binding to the individual monomers [4] assuming that binding sites on each monomer are equal and independent [21].

Using *R* described in Equation (3), the *K_D_* calculation can be re-written into Equation (4) with the known initial concentration of glycan ligand [*L*_0_] and protein [*P*_0_], adapted from Sun et al. [5]:(3)$$R=PL_{equil}∕P_{equil}=\underset{n}{\sum }A(PL^{n+})∕n∕\underset{n}{\sum }A(P^{n+})∕n$$
(4)$$K_{D\;}=\frac{L_{0}-\frac{RP_{0}}{1+R}}{R}$$

### 2.5. Titration Measurement of Glycan Binding on P Dimers

For the carbohydrate-binding experiments, it was essential to keep the initial protein concentrations and pH value constant during the analysis of the protein-carbohydrate solutions. While acidification occurs in any ESI droplet, the remaining solution in the capillary changes pH over time. Therefore, accumulated spectra were obtained within 10 min after the voltages were applied to minimize the influence of pH changes during the measurements. The LCT mass spectrometer with only a short hexapole prior to the ToF analyser reduces potential activation and hence in-source dissociation (ISD). Three different glycan ligands including ligands of interest (HBGA B and A) and non-binding control (Gb4) were measured in a range of concentrations to determine the *K_D_* of the first glycan binding event to P dimers (Table 1).

### 2.6. Ion Mobility Mass Spectrometry (IMMS)

IMMS experiments on the GII.4 Saga P dimers were performed in a Synapt G2-S instrument modified with a linear drift-tube ion-mobility cell operating with He drift gas under 2.4 mbar pressure. Samples with 3 µM P dimer protein and 0, 100, 200, and 500 µM of the HBGA B tetrasaccharide or Gb4, respectively, were prepared, thoroughly homogenized, then loaded onto a Pd/Pt-coated borosilicate glass capillary. The sample was introduced by nano-ESI with the following instrumental parameters: capillary voltage, 1200 V; cone voltage, 15 V; source offset, 15 V; source gas flow, 0.0 mL/min; backing pressure, 0.1 bar; source temperature, 30 °C. These settings ensured soft ionization conditions that allowed the preservation of the native protein structure. The arrival time distribution curves shown later on in Figure 5 were recorded at a He cell DC of 35 V and a bias of 55 V. Each experiment was performed at least twice to address reproducibility.

## 3. Results

### 3.1. Glycan Clustering on Norovirus P Dimers

Here, the wildtype and deamidated GII.4 Saga P dimers are used as positive and negative protein-binding control [4], respectively. The tetrasaccharide of HBGA B type 1 is employed as a glycan known to bind to the wildtype and Gb4 as an all galactose glycan non-binder [9,10]. Thereby, the applicability of the MS approach to investigate binding equilibria in presence of a reference protein is verified. As can be seen in Figure 1A, the reference protein cytochrome c (cyt c) picks up similar amounts of clustered glycans in both spectra indicating similar spray conditions. Surprisingly, the wildtype and deamidated P dimers also reveal a comparable pattern of glycan attachment although a reduced amount of glycans is expected to stick to the deamidated protein. After correcting for clustering to the reference protein, even more glycans are supposedly specifically bound to the deamidated P dimer, which serves as a low- or non-binding control, than to the wildtype. Moreover, occupancy is much higher than would be expected based on the K_D_s for the wildtype determined by NMR (12 mM for B tetrasaccharide [18]), suggesting an intrinsic issue with the measurement approach. Additionally, the non-binding glycan Gb4 shows a pattern similar to HBGA B after correction (Figure 1B). but with more glycan attached to the wildtype protein.

Protein NMR experiments recently demonstrated that MNV P dimers do not bind to HBGAs [18]. Moreover, MNV shows altered dimerization properties and presents a large fraction of P monomers at neutral pH. While correction is not possible due to the insufficient quality of the reference protein signal (see Appendix A), it is evident that similar glycan clustering is observed for the P monomer and dimer. Glycan binding requires a dimeric protein; therefore, these interactions between MNV P monomers [22] and glycans have to be unspecific in line with the study by Creutznacher et al. [18]. Additional support for largely unspecific interactions stems from a mutated hNoV GII.4 MI001 P dimer. Here, the supposed glycan-binding pocket is mutated resulting in altered dimerization behavior; hence, the P monomer and dimer signals. The corrected data shows equal clustering behavior for HBGA B, A and 3′-sialyllactose (GM3) on the monomer and dimer (Appendix A). Interestingly, Gb4 shows basically no binding after correction in this case. Furthermore, the negatively charged GM3 shows similar patterns compared to the neutral HBGA glycans, both on the monomer and dimer resulting in similar occupancy after correction. This suggests that charged glycans also suffer from similar problems in the direct MS approach. Notably, correction for clustering based on the P monomer also results in no specific binding to the P dimer for the other three glycans. This suggests that cyt c is not a suitable reference and that glycan clustering does depend on the biophysical properties of the proteins. Therefore, we compare several potential reference proteins, that are commercially available and exhibit distinct properties.

### 3.2. Glycan Clustering to Various Reference Proteins

The ratio, *R,* between the free and ligand-associated reference protein is used to eliminate non-specific clustering to the target protein. In total, eight different reference candidates differing in mass, size and structural composition display very different *R* values (Table 1). For example, the interaction of carbohydrates with the Saga P dimer and four reference proteins (cyt c, GFP, CA, and ADH) is shown (Figure 2). The data illustrated similar binding patterns of HBGA B to Saga P dimers but vastly different glycan clustering to the reference proteins under identical measurement conditions. The dimeric ADH displays the strongest ligand clustering with ADH–ligand (1:1) becoming the base peak similar to the P dimer. CA and GFP show similar patterns with slightly less clustering, whereas cyt c presents a unique profile with lower clustering ratios and a non-Gaussian charge state distribution.

**Table 1 life-11-00554-t001:** Glycan clustering ratio *R* and K_D_ of HBGA B interaction with P dimers analysed by native MS. Data correction is based on the reference protein method. n.a.–not available.

Ref. Protein	Mass /kDa	PDB ID	Secondary Structure Composition		Glycan-Clustering Ratio *R* at 500 µM			K_D_ for HBGA B /mM
			% β-sheet	% α-helix	HBGA B	Gb4	HBGA A	
Ubq	8.9	1UBQ	24.7	20.8	0.25 ± 0.00	0.09 ± 0.03	0.10 ± 0.05	0.20 ± 0.13
Cyt c	13.2	2N3B	0	53.8	0.35 ± 0.03	0.45 ± 0.02	0.41 ± 0.09	0.16 ± 0.05
Myo	17.0	1AZI	0	79.2	0.36 ± 0.05	0.37 ± 0.06	0.34 ± 0.03	0.25 ± 0.10
GFP	26.8	1BFP	55.0	18.5	0.67 ± 0.02	0.80 ± 0.03	0.86 ± 0.04	0.90 ± 0.35
CA	29.1	1V9E	35.8	15.8	0.75 ± 0.07	0.75 ± 0.03	0.74 ± 0.07	0.40 ± 0.10
apo-TFF	79.6	2HAU	15.7	34.5	0.82 ± 0.03	0.80 ± 0.05	0.79 ± 0.03	n.a.
LDH	146.8	3H3F	17.1	46.9	1.02 ± 0.04	0.91±0.06	n.a.	n.a.
ADH	147.0	4W6Z	34.8	28.4	1.45 ± 0.11	1.26 ± 0.03	1.32 ± 0.10	24.00 ± 16.00
SAGA P dimers	68.1	4X7C	32.5	6.6	1.55 ± 0.05	1.40 ± 0.21	1.44 ± 0.15	n.a.

In line with the observed discrepancies in clustering ratio *R*, the resulting K_D_ values for the same protein-ligand interaction vary strongly between a few 100 µM and 24 mM depending on the choice of the reference protein. Notably, the K_D_ obtained with ADH as reference protein is in line with the expected value of 12 mM from NMR data [18] but is also barely detectable (Appendix A). Differences in *R* and K_D_ are also consistent when titrating five glycan concentrations (Appendix A). When the clustering ratio *R* is plotted over the glycan concentration (Figure 3), all reference proteins show the expected proportionality of *R* and glycan concentration for the three tested glycans but with different slopes. The plots clearly reveal the strongest clustering to ADH in all conditions. On the other hand, the small-sized proteins (Myo, cyt c and Ubq, approx. 17, 13 and 9 kDa, respectively) stick out with much lower clustering. The remaining proteins display intermediate *R* values. This may suggest an influence of molecular weight on the reference protein despite other reports [5,11]. While all plots fit linear regression, the plots based on the reciprocal *R* value reveal that clustering to Ubq is not linearly dependent on glycan concentration (Figure 3, grey line). Therefore, Ubq is excluded from the following analysis.

### 3.3. Biophysical Properties Influence the Glycan Clustering

Ten different protein characteristics that could have affected glycan clustering are plotted against the *R* values of the remaining 7 reference protein candidates to dissect possible correlations: the absolute amount of α-helix or β-sheet in kDa, number of total charged residues, total number of positively or negatively charged residues, isoelectric point (pI), *m/z* values, molecular weight, solvent accessible surface area (SASA), and collision cross-section calculated from trajectory method (CCS TJM) (Figure 4, Appendix A). Glycan protein associations are often mediated by CH–π interactions with tryptophan residues. Such interactions are expected to be weak in the gas phase, hence number of tryptophans is not included. Notably, the SAGA P dimer contains only 8 tryptophans, which is below average for its size, clearly showing that tryptophan content cannot explain the huge clustering ratio for the P dimer. *R²* is used to assess the quality of the linear fits. The only parameter providing strong correlations at all ligand concentrations is the β-sheet amount of the reference proteins (Figure 4), whereby the β-sheet amount and glycan clustering grow proportionally. The absolute share of β-sheets in a protein is also connected to the molecular weight. The larger the protein, the more amino acids engage within β-sheets, which is the likely explanation for the apparent impact of protein size. Notably, the amount of α-helix shows no anti-proportional or any other correlation, indicating that these contribute marginally to glycan clustering. Some of the parameters have *R²* values between 0.7 and 0.9, i.e., significantly lower than the β-sheet amount; however, a contribution to overall glycan clustering cannot be excluded for SASA, charged residues, molecular weight and *m/z*. CCS TJM and pI on the other hand are below 0.5, like the α-helices, and clearly do not contribute.

### 3.4. Ion Mobility on P Dimer Glycan Interactions

Native ion mobility MS (IMMS) can reveal conformational changes in proteins, e.g., upon ligand binding, and has also been used to investigate P dimer glycan binding [9]. In IMMS, changes in arrival time above 3–5% are generally considered significant, which are not expected here based on from HDX-MS and NMR [4]. Hence, the binding of a small ligand like HBGA B should only cause marginal changes. However, in a previous report, a strong size increase upon glycan addition to SAGA P dimers was observed in IMMS [9]. We hypothesize that interaction of the glycans with the β-sheets lead to the melting of these structures in the gas phase, which could be observed in IMMS. In Figure 5, the arrival-time distributions are shown for the wildtype GII.4 Saga P dimer together with its complexes with HGBA B (Figure 5A) and Gb4 (Figure 5B) for all detectable stoichiometries, as well as the respective single complexes of 1 P dimer (0–3 HGBA B (Figure 5C)) and 1 P dimer: (0–3 Gb4 1 molecules (Figure 5D)). A small difference in the mean values and the peak widths is observed among the panels A and B, and C and D in Figure 5, which is due to changes in environmental conditions, such as temperature, and the slightly different tuning parameters used to optimize the signal. To ensure that no conformational changes are induced by the different conditions, the rotationally averaged collision cross-section (CCS) for the pure P dimer is calculated: in A and C, the CCS is 3928 ± 14 Å^2^, while in B and D, the CCS is 3867 ± 12 Å^2^. The difference between these two CCS values is ~1.6%, which is well below the significance threshold of the instrument, and is normal for experiments with large molecules. It could be seen that the arrival-time distribution curves preserve their near-Gaussian, single-peak profile at all P dimers. The ligand stoichiometries suggest no major conformational changes upon complex formation and also no evidence of structural destabilization in the gas phase. Even though the arrival-time distributions for all complexes (Figure 5A,B) do not seem to follow an obvious trend, the difference between the distribution of the P dimer (black traces) and the sample with 500 µM of the respective ligands (green traces) suggest that at high-ligand concentration, the relative concentration of the free P dimer decreased. As can be seen in the distributions of selected complexes (Figure 5C,D), with increasing number of bound ligands, the arrival-time distribution curves shift slightly towards higher values. The data clearly show that the binding- and the non-binding ligands trigger the same behaviour, which we largely attribute to clustering.

## 4. Discussion

Glycan clustering was previously observed in native MS experiments [5,9,10,11]. It occurs during ionization when solvent evaporates and the ions transfer to the gas phase. Due to an excess of ligand, free glycans statistically end up in the same droplet and dry down onto the protein next to specifically interacting glycans. The addition of a reference protein enables the determination of non-specific ligand clustering and hence correction to allow direct MS analysis of the binding occupancy and K_D_. Therefore, this method has been widely used [12,13,14,23] to study protein–ligand interactions. Glycans pose a specific problem as the interactions are often of low affinity in the mM range and require vast ligand excess to occupy binding sites.

Using deamidated GII.4 Saga P dimers as well as MNV and MI001 P monomers as negative controls for P dimer-glycan interaction, we reveal inherent problems with the direct MS approach for employing reference proteins. This is further corroborated by complementary experiments using the non-binding all galactose glycan Gb4. The binding incompetent monomers show the same extent of glycan association as the respective P dimers do. Of note, in these cases, binding to the P dimers is also not expected due to mutations and MNV P dimers being unable to bind glycans at all [18]. The results indicate that these problems are not limited to neutral glycans but also occurre for sialylated glycans. Nevertheless, for some glycans, MS yields even higher affinities for deamidated GII.4 Saga P dimers than for the wildtype, contradicting results from NMR and HDX-MS, which showed increased flexibility in the deamidated P dimers, suggesting that the structure affects glycan clustering. Here, we re-examine the degree of glycan attachment to different reference proteins to elucidate its origin and general suitability in native MS.

We select eight reference proteins differing in properties. The clustering on ADH appears similar to the glycan distribution on the Saga P dimers while cyt c only presents a small amount of clustering. This further confirms that glycan clustering is influenced by the protein’s physicochemical properties. While most proteins show a linear correlation between clustering ratio and glycan concentration, Ubq behaves differently. In general, folded proteins are thought to ionize via the charged residue model (CRM). However, the non-linear glycan clustering behaviour of Ubq hints at ionization following another model, as previously suggested by MD simulations. According to these simulations small proteins can also be ionized through the ion evaporation model (IEM) [24], which would explain the peculiar behaviour of Ubq and may also play a role for other small proteins. The cyt c clustering patterns have a non-Gaussian character, which could be caused by ISD or varying ESI efficiencies. In contrast to Han et al., 2013 and 2018 [9,11], these results indicate an unexpectedly different carbohydrate-clustering propensity, resulting in K_D_s spanning two orders of magnitude. The question arises: which biophysical characteristics caused this effect?

Previous work suggested that the mass and size of the reference proteins did not affect the correction procedure significantly [5,9]. Since most carbons in glycans carry hydroxyl groups, glycans have a much higher hydrogen bonding capacity than other small molecules, which could affect the interaction with the protein surface during ESI. Therefore, inspection of the structure of the selected reference proteins reveals distinct ratios of secondary structure elements. Various criteria are tested for correlation to the clustering ratio with the strongest correlation observed for β-sheet contents. We hypothesize that the net-like hydrogen bonding pattern in the β-sheets favours interaction with the glycans in contrast to α-helical structures. This could imply that intercalation occurres during ionization when glycans attache to the protein surface. Furthermore, β-sheets are more labile during the ESI process [25], which could facilitate glycan intercalation. Notably, a reasonable correlation is also observed for molecular weight and SASA, which could be related to the higher probability of containing a significant amount of β-sheets with increasing size.

The direct MS approach with a reference protein is therefore not well suited for studying low-affinity glycan binding. It was originally developed for higher-affinity interactions where agreement in K_D_ to other methods was observed [5]. It has also proven invaluable for other ligand types [8,23,26]. IMMS demonstrates consistent results for HBGA B and Gb4, implying similar glycan clustering. No major changes in arrival-time distributions are observed upon glycan addition in accordance with small ligands being added, and little structural changes were observed in NMR and HDX-MS [4]. This contrasts with a previous report [11] where the experimental and data acquisition parameters differed. Conformational changes were reported for glycan charge states 17+ and 18+ up to 800 µM, and we have also checked HBGA B and the 17+ charge state at 800 µM but find no indication of conformational changes. Notably, we use a modified instrument containing a drift tube as opposed to the travelling-wave ion-mobility device employed by the other group, which could have caused overactivation and unfolding. Another explanation could be an isobaric contaminant with a larger gas phase structure. However, in our case, we are certain that isobaric contaminants are not present and overactivation of the structures does not occur due to the single near-Gaussian shape of the IMMS peaks.

Overall, our results suggest that reference proteins with similar properties to the protein of interest should be used. Moreover, small proteins that could be affected by IEM upon ESI should be avoided. This explains some of the observed K_D_ discrepancies in the literature. We had previously used cyt c, both small and mostly α-helical, which overestimated the binding affinity [11,12]. Others used a small-sized monoclonal antibody single-chain fragment (scFv 26 kDa, [5,9,11]), which mostly consisted of β-sheets. While the latter was much better suited, our data show that protein dynamics, as in the deamidated P dimer, further influence clustering [4,27].

The introduction of a non-binding ligand control is an additional assessment parameter to confirm the specific binding for low-affinity ligands, which has also been performed for glycan mimetics [13]. After an ADH based protein clustering correction, the non-binder Gb4 expressed no specific binding in any listed glycan concentration, and HBGA B shows a single binding event at 500 µM concentration (compare to the stoichiometry information (Appendix A)). The calculated K_D_ (24 mM) is in accordance with the result from NMR (K_D_: HBGA B-tetrasaccharide type 1 12 mM, B trisaccharide 6.7 mM, fucose 22 mM) [4,18]. Hence, an appropriate reference candidate is crucial for the K_D_ determination for low-affinity glycans.

## 5. Conclusions

In low-affinity glycan binding studies, the K_D_ calculation is heavily dependent on the degree of glycan clustering on the reference protein. The protein’s structure and dynamics seem to heavily influence the degree of glycan clustering. Therefore, the quantification of direct binding affinity requires the careful selection of a reference protein with similar β-sheet content and even similar structural dynamics to obtain accurate binding occupancy and affinities. The recent introduction of submicron emitters could be a way to reduce or circumvent the problem [28].

## Data Availability

All data is available from the corresponding author upon request.

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
