# Peer review of "Protein Secondary Structure Affects Glycan Clustering in Native Mass Spectrometry"

_life, 2021, doi:10.3390/life11060554_

Round 1

Reviewer 1 Report

Yan et al. analyze glycan clustering on proteins during native mass spectrometry measurements. Specifically, the analyze unspecific glycan binding occurring during the electrospray ionization process. Unspecific interactions between proteins and their ligands have been reported previously and correction of the mass spectra by addition of reference proteins was suggested. Yan et al. now show that correction of mass spectra strongly depends on the biophysical properties of the reference proteins explaining contradictory results from previous studies. Employing different glycan binding proteins, non-binding variants as control, several reference proteins as well as various glycans (ligands) they extract KD values from their measurements and link the obtained results with various biophysical properties of the reference proteins. They conclude that beta-sheet content (and to some degree molecular weight and SASA) is an important factor affecting suitability of reference proteins; the authors therefore suggest the use reference proteins with similar properties for correction of mass spectra.

The study of unspecific interactions occurring during native mass spectrometry is an important issue that should be explored more frequently. The authors make a first contribution exploring unspecific binding of glycans, which represent only one class of ligands that unspecifically attach to proteins during electrospray ionization. The study is overall thoroughly conceived and performed. The results are presented clearly and the manuscript is well written. I suggest publication of the manuscript after addressing few minor comments.

  1. Abbreviations should be checked. Most abbreviations that are introduced in the abstract are introduced again in the main text, however, some are missing (e.g. NMR). Abbreviation for Cytochrome C is not consistent (compare Figure 1 and Figure 2). Abbreviation ESI is introduced but sometimes not used (line 198).

  1. Lines 185-187. Mass spectra were acquired within 10 minutes after voltages were applied to avoid changes in pH during the ESI process. During ESI the solvent evaporates and the number of protons increases in the droplets. This is provoking pH changes. These changes are occurring in all ESI droplets independently on the time frame of measurements. The authors should clarify this statement.

  1. The authors use wild-type GII.4 Saga P dimers as well as a deamidated variant thereof. In the beginning and in the end of the manuscript the authors refer to wild-type while in the results section they mostly refer to non-deamidated Saga P dimer. This is confusing. I suggest that the authors use the terms wild-type and deamidated throughout (instead of deamidated and non-deamidated).

  1. Figure 2. Mass spectra are shifted. This is slightly confusing because charge state distributions are also shifted which gives the impression that charge states are different in the different mass spectra. I suggest to present all spectra directly above each other or to add auxiliary lines for better comparison.

  1. The authors should stronger emphasize the rationale behind the IMMS experiments. It is not entirely clear why these experiments were performed.

  1. Line 316-318: (A), (B), (C) and (D). Please add the information that these are Figure panels.

  1. Figure 5, panels C and D: m/z is given for the protein and protein adducts. I suggest to change this to P, P+1L, P+2L, P+3L or similar. This would be easier to follow.

  1. How many replicates were performed for IMMS measurements? Are these small changes reproducible?

  1. Glycan binding is often attributed to specific amino acids (e.g. aromatic amino acids). It might be worth checking the content of these amino acids on the surface of the reference proteins.

Additional comment: Please check affiliations of authors. Affiliation 6 is not used!

Author Response

Yan et al. analyze glycan clustering on proteins during native mass spectrometry measurements. Specifically, the analyze unspecific glycan binding occurring during the electrospray ionization process. Unspecific interactions between proteins and their ligands have been reported previously and correction of the mass spectra by addition of reference proteins was suggested. Yan et al. now show that correction of mass spectra strongly depends on the biophysical properties of the reference proteins explaining contradictory results from previous studies. Employing different glycan binding proteins, non-binding variants as control, several reference proteins as well as various glycans (ligands) they extract KD values from their measurements and link the obtained results with various biophysical properties of the reference proteins. They conclude that beta-sheet content (and to some degree molecular weight and SASA) is an important factor affecting suitability of reference proteins; the authors therefore suggest the use reference proteins with similar properties for correction of mass spectra.

The study of unspecific interactions occurring during native mass spectrometry is an important issue that should be explored more frequently. The authors make a first contribution exploring unspecific binding of glycans, which represent only one class of ligands that unspecifically attach to proteins during electrospray ionization. The study is overall thoroughly conceived and performed. The results are presented clearly and the manuscript is well written. I suggest publication of the manuscript after addressing few minor comments.

  1. Abbreviations should be checked. Most abbreviations that are introduced in the abstract are introduced again in the main text, however, some are missing (e.g. NMR). Abbreviation for Cytochrome C is not consistent (compare Figure 1 and Figure 2). Abbreviation ESI is introduced but sometimes not used (line 198).

Additional abbreviations are now introduced and cytochrome c abbreviation checked for consistency.

2. Lines 185-187. Mass spectra were acquired within 10 minutes after voltages were applied to avoid changes in pH during the ESI process. During ESI the solvent evaporates and the number of protons increases in the droplets. This is provoking pH changes. These changes are occurring in all ESI droplets independently on the time frame of measurements. The authors should clarify this statement.

We thank the reviewer for pointing this out correctly and added further explanation in line 186: “While acidification occurs in any ESI droplet, the remaining solution in the capillary changes pH over time.”

3. The authors use wild-type GII.4 Saga P dimers as well as a deamidated variant thereof. In the beginning and in the end of the manuscript the authors refer to wild-type while in the results section they mostly refer to non-deamidated Saga P dimer. This is confusing. I suggest that the authors use the terms wild-type and deamidated throughout (instead of deamidated and non-deamidated).

We agree and changed non-deamidated to wild-type throughout text and figures.

4. Figure 2. Mass spectra are shifted. This is slightly confusing because charge state distributions are also shifted which gives the impression that charge states are different in the different mass spectra. I suggest to present all spectra directly above each other or to add auxiliary lines for better comparison.

This figure has been aligned to avoid confusion.

5. The authors should stronger emphasize the rationale behind the IMMS experiments. It is not entirely clear why these experiments were performed.

The IMMS experiments were performed as others previously suggested that glycan addition results in conformational changes in SAGA P dimers. We had the hypothesis that glycan clustering at high concentrations could maybe result in melting of the beta sheets, which would explain an unusually high increase in collision cross section for addition of small ligands, now stated in lines 334ff. This hypothesis could not be confirmed as we only observed marginal size increase upon glycan addition as would normally be expected for small ligands.

6. Line 316-318: (A), (B), (C) and (D). Please add the information that these are Figure panels.

Done, now line 338.

7. Figure 5, panels C and D: m/z is given for the protein and protein adducts. I suggest to change this to P, P+1L, P+2L, P+3L or similar. This would be easier to follow.

We added stoichiometry information to panels C and D in addition to the m/z.

8. How many replicates were performed for IMMS measurements? Are these small changes reproducible?

See line 205, experiments were performed at least twice to ensure reproducibility. The results are qualitatively consistent between glycans. In the results section, we moreover state that these small changes are below the significance threshold clearly arguing against large structural changes.

9. Glycan binding is often attributed to specific amino acids (e.g. aromatic amino acids). It might be worth checking the content of these amino acids on the surface of the reference proteins.

We thank the reviewer for this thoughtful suggestion. For glycan binding, CH-pi interactions to tryptophan are most relevant. We expect such interactions to be very weak in the gas phase as opposed to hydrogen bonds or electrostatic interactions and hence had not included number of tryptophans in the analysis. For the reference proteins, we observe a poor correlation as for the charged residues, which is likely a remnant of the mass/beta sheet increase going alongside. What clearly argues against tryptophan influence, is the fact that SAGA P dimer has a particularly low amount of tryptophan not in line with the very high clustering ratio. We added a short clarification in the results section in lines 307ff.

Additional comment: Please check affiliations of authors. Affiliation 6 is not used!

Thanks for spotting. The affiliation has now been linked to the corresponding author.

Reviewer 2 Report

Yan et al, in this study have shown that data emerging from MS based binding studies must be interpreted cautiously. Using native and ion mobility MS they demonstrate that the mass fraction of β-sheet has a strong influence on the degree of glycan clustering.

They emphasize that reference proteins need to be considered to match structural properties of the protein of interest for glycan interaction studies. Furthermore, influence of structural dynamics are also suggested.

The reviewer finds this study of interest to the journal. Overall it has been well written, with data supporting their arguments and well illustrated.

A few issues that the authors should address are:

The study focuses mostly on the biophysical properties of the protein, there is lack of information or data for non specific glycan clustering that can be contributed when involving glycans that have potential charge for example in the case of sulphated or siaylated structures.

The negative control used here Gb4 is neutral similar to HGBA ligads. It would be of interest to observe and test against a glycan such as GM1.

Do the charge state of the complex influence the degree of glycan clustering? there seems to some observable variation in Figure 1 and 2.

In Page 14, line 419, the authors say "No major changes in arrival time distributions are observed upon glycan addition in accordance with small ligands being added, and little structural changes observed in NMR and HDX-MS [4]. This is in contrast to a previous report [11], where the experimental and data acquisition parameters differed." It would be of interest to the readers to describe and discuss in more detail the contrasting parameters that resulted in different results.

Author Response

Yan et al, in this study have shown that data emerging from MS based binding studies must be interpreted cautiously. Using native and ion mobility MS they demonstrate that the mass fraction of β-sheet has a strong influence on the degree of glycan clustering.

They emphasize that reference proteins need to be considered to match structural properties of the protein of interest for glycan interaction studies. Furthermore, influence of structural dynamics are also suggested.

The reviewer finds this study of interest to the journal. Overall it has been well written, with data supporting their arguments and well illustrated.

A few issues that the authors should address are:

The study focuses mostly on the biophysical properties of the protein, there is lack of information or data for non specific glycan clustering that can be contributed when involving glycans that have potential charge for example in the case of sulphated or siaylated structures.

This is a valuable suggestion, we actually show data for the sialyted glycan GM3 clustering to the mutated MI001 P dimer in Fig. S2. While we have not systematically investigated charged glycans, the data suggests similar clustering behaviour as for the neutral glycans. This is evident from similar ratios of GM3 on monomers and dimers. Additionally, GM3 bound after correction compares well to the amount of neutral glycans. We now clearly point that out in results and discussion in lines 244ff and 381f.

The negative control used here Gb4 is neutral similar to HGBA ligads. It would be of interest to observe and test against a glycan such as GM1.

See above.

Do the charge state of the complex influence the degree of glycan clustering? there seems to some observable variation in Figure 1 and 2.

Well spotted, sometimes there seems to be a charge state dependence and in other cases not. We attribute this to spray variation and associated differences in activation. Also, sometimes the P dimer series starts at 19+ in other cases at 17+. This is not consistent or quantifiable, thus we refrained from pointing this out.

In Page 14, line 419, the authors say "No major changes in arrival time distributions are observed upon glycan addition in accordance with small ligands being added, and little structural changes observed in NMR and HDX-MS [4]. This is in contrast to a previous report [11], where the experimental and data acquisition parameters differed." It would be of interest to the readers to describe and discuss in more detail the contrasting parameters that resulted in different results.

It is difficult to say what caused the exact differences. We suspect overactivation or some contamination in the other report but cannot say for sure. We expanded the discussion slightly to reflect, which parameters were consistent between the experiments (lines 425ff).